# Invasive Pulmonary Aspergillosis

**DOI:** 10.3390/jof9020131

**Published:** 2023-01-17

**Authors:** Marie-Pierre Ledoux, Raoul Herbrecht

**Affiliations:** Department of Hematology, Institut de Cancérologie de Strasbourg Europe (ICANS), 67033 Strasbourg, France

**Keywords:** leukemia, hematopoietic stem cell transplantation, solid organ transplantation, neutropenia, galactomannan, voriconazole, posaconazole, isavuconazole, liposomal amphotericin B, review

## Abstract

Invasive pulmonary aspergillosis is growing in incidence, as patients at risk are growing in diversity. Outside the classical context of neutropenia, new risk factors are emerging or newly identified, such as new anticancer drugs, viral pneumonias and hepatic dysfunctions. Clinical signs remain unspecific in these populations and the diagnostic work-up has considerably expanded. Computed tomography is key to assess the pulmonary lesions of aspergillosis, whose various features must be acknowledged. Positron-emission tomography can bring additional information for diagnosis and follow-up. The mycological argument for diagnosis is rarely fully conclusive, as biopsy from a sterile site is challenging in most clinical contexts. In patients with a risk and suggestive radiological findings, probable invasive aspergillosis is diagnosed through blood and bronchoalveolar lavage fluid samples by detecting galactomannan or DNA, or by direct microscopy and culture for the latter. Diagnosis is considered possible with mold infection in lack of mycological criterion. Nevertheless, the therapeutic decision should not be hindered by these research-oriented categories, that have been completed by better adapted ones in specific settings. Survival has been improved over the past decades with the development of relevant antifungals, including lipid formulations of amphotericin B and new azoles. New antifungals, including first-in-class molecules, are awaited.

## 1. Introduction

*Aspergillus* is a saprophytic and ubiquitous mold, its spores being found both in indoor and outdoor air. They arise from various sources such as soil, decomposing plant matter, plant and flowers, food, household dust or building materials [1,2]. High levels of spore aerosolization, resulting in outbreaks of infections, have been reported under specific local geoclimatic conditions, after construction works, or malfunction of air filtration systems [3,4,5]. Fungal genus Aspergillus is divided into subgenera, sections and species [6]. The most clinically significant section is Fumigati, comprising of *A. fumigatus*, *A. lentulus* and *A. udagawae* among others [7]. Most relevant species outside the Fumigati section are *A. flavus*, *A. nidulans*, *A. terreus* and *A. niger* [8]. *A. fumigatus* is the most frequently involved in human diseases in reports from America and Europe, while *A. flavus* is gaining prevalence in some Asian countries, notably in rhino-orbito-cerebral infections [9]. In patients’ care, daily practice requires the identification of the strain to the level of section, whereas further identification, to the species level, is possible through the use of molecular testing [10].

In appropriate humidity and temperature conditions, Aspergillus species grow into hyphae and eventually produce conidia, an airborne form of infectious potential. The contamination with this mold happens therefore mostly through airways, with lungs being the first site of invasive aspergillosis (IA) in terms of frequency. Although all patients are exposed to spores, only severely immunocompromised ones develop IA. Aspergillus is not a normal component of the lung mycobiome, as it is supposed to be eliminated by physiological host response [11,12]. The first line of this host defense is represented by innate immunity: neutrophils have a pivotal role against the mold through various processes including phagocytosis, oxidative burst and the more recently described neutrophil extracellular traps [13]. These first line actors trigger adaptative immunity, resulting in a T-helper cell TH1 response associated with increased production of interferon, IL-2 and IL-12 and stimulation of effector cells. On the other hand, a level of tolerance is TH2-mediated through IL-4 and IL-10 [12]. The normal host response results in a clearance of the conidia before further development and without excessive inflammatory reaction.

The degree of failure of the host response explains the wide scope of clinical forms of *Aspergillus* colonization or infections [14]. Immune hyperreactivity can be found in asthma and cystic fibrosis, resulting in allergic bronchopulmonary aspergillosis (ABPA). Chronic forms of pulmonary aspergillosis, including aspergilloma and chronic pulmonary aspergillosis (CPA), are characterized by the absence of angioinvasion and hyphae confined to a pre-existing lung cavity. They are therefore associated with pre-existing structural lung disease and only minor immunological defect and are seen in patients with chronic obstructive pulmonary disease (COPD) or sarcoidosis. When a certain degree of immunosuppression is added, for instance with corticosteroid therapy in COPD, hyphae can appear in tissue, resulting in an invasive form of CPA, or invasive pulmonary aspergillosis without angioinvasion. Chronic necrotizing aspergillosis or subacute invasive aspergillosis is a further step in this continuum. On the most immunosuppression-associated end of this continuum, IA is best described in neutropenic hosts, where it is classically characterized by hyphae in the tissue and images of angioinvasion, coagulative necrosis and hemorrhagic infarction.

## 2. Patients

The ability for *Aspergillus* to develop into an IA in a given patient depends on a balance between exposure and host defenses. Some conditions have long been known as important risk factors for IA, and others are still being uncovered, due to new diseases or new risk emerging from evolving therapies of already well-known diseases. On the whole, annual incidence of IA is estimated > 300,000, among ≈10 million patients at risk, in comparison with ≈3,000,000 patients sharing the global burden of CPA, ≈4,800,000 ABPA in asthma and >6000 ABPA in cystic fibrosis [15]. A compilation of two large phase 3 clinical trials showed that >85% of IA occurred in patients with hematological malignancy, or in recipients of allogenic hematopoietic stem cell (alloSCT) or solid organ transplantation [16,17].

### 2.1. Acute Myeloid Leukemias and Myelodysplastic Syndromes

Patients presenting with deep and prolonged neutropenia in the course of acute myeloid leukemia (AML) or myelodysplastic syndromes (MDS) are the paragon of risk for IA. Incidence in this population can reach 24% in some series [18]. Most of the diagnosis occur during the aplasia following the induction therapy, although some occur during the consolidation courses [18]. Relapsed and refractory diseases are associated with an even higher risk for invasive mold diseases (IMD).

In spite of a profound neutropenia, MDS patients who are treated with hypomethylating agents rather than chemotherapy induction are not that prone to IA, indicating that the neutrophils rate probably does not account for the whole risk in AML patients [19]. Moreover, AML patients who are treated with the intermediate intensity scheme associating azacytidine and the apoptosis-restoring drug venetoclax seem to show intermediate risk of IMD, mostly IA, of between 5.1% and 19%, but data are still scarce on that point [20].

AML and intensively treated MDS are well-defined indications for mold-active prophylaxis, based on posaconazole or voriconazole, advised during the induction course depending on the local epidemiology [21]. Nevertheless, optimal management can be difficult if a target therapy is used for the treatment of the hematological malignancy, as many of them show drug–drug interactions with azoles: for instance, venetoclax, FLT3-inhibitors or ivosidenib (IDH1 inhibitor) require dose-adjustment in case of concomitant azole [22,23,24].

Of note, IA is usually associated with better outcome in the setting of AML and a chemotherapy-induced neutropenia thanks to the reversibility of the main risk factor, i.e., neutropenia [25].

### 2.2. Other Hematological Malignancies

Although their management includes high intensity chemotherapy and high dose corticosteroids, acute lymphoblastic leukemias (ALL) are associated with a lower risk of IA than AML. Authors relate an incidence of between 2.2% and 15.4% of IMD, IA being the most frequent [26,27]. Interestingly, ALL patients seem less prone to pulmonary forms of IA, with more rhino-sinusal infections [28]. Relapsed or refractory disease is associated with a higher risk of IA, with an incidence of 23.5% compared to 6.2% in a pediatric series and 28.5% vs 5.6% in adult patients: of note, the chemotherapy in this setting tends to resemble that of AML [27,29]. Little is known yet about newer target drugs for ALL: bivalent antibodies and conjugated antibodies do not seem to directly worsen the risk for fungal infections but may result in neutropenia, which has been associated in this setting with nodular pneumonia, suggesting IA, rather than other infections [30,31,32].

Lymphoproliferative disorders were historically less associated with a high risk of IMD and especially IA. Recently though, stress has been put on a series of cases among patients receiving target therapies [33,34]. A series of 33 IFD comprising 27 IA has been described in patients under ibrutinib, a Bruton-tyrosine kinase inhibitor aimed at targeting B cell but showing off-target impact on neutrophilic functions [35,36]. A proportion as high as 40% of these patients showed cerebral involvement of their IA. For a hint on incidence, a longitudinal study on patients under ibrutinib shows 43 severe infections in 378 patients, 18.6% of them being IA [37].

The evolution of treatment of multiple myeloma tends to narrow the role of chemotherapy and give overwhelming role to targeted therapy: associations of proteasome inhibitors, immunomodulators and monoclonal antibodies are pivotal in the improvement of myeloma patients’ survival, but the crucial role of dexamethasone and intensification with high-dose melphalan remains. Telling the different factors apart among the evolving therapies is difficult, but some authors observed a rise in incidence of IFD in myeloma patients: from 0.5% in 2006 to 5.6% in 2016 in an Italian cohort [38,39]. However, IA seems to account for a lower proportion of the 3.4–3.5% IFD observed in recent myeloma cohorts, with more frequent pneumocystis pneumonias and candidemias [40,41].

### 2.3. Allogeneic Hematopoietic Stem Cell Transplantations

Patients undergoing alloSCT are a historically well-known group at risk for IA, from the early neutropenic period to the following acute then chronic graft-versus-host disease (GVHD). An incidence of 15% for invasive fungal diseases (IFD) was described in a seminal publication in 1997, most of them being IA [42]. Only a small proportion (14%) happened during the early neutropenic period, suggesting the various mechanisms of alloSCT-associated risk. In a more recent publication, those factors were analyzed according to the post-graft delay: IA occurring before day 40 were associated with lack of engraftment, IA occurring in a late period between day 40 and day 100 were associated with grade > 2 acute GVHD, and very late, after day 100, IA were associated with grade > 2 GVHD, relapse and secondary neutropenia [43]. Another publication has investigated the impact of donor source on the risk of fungal infections and showed that the risk was higher in mismatched alloSCT compared to matched unrelated donors (16% vs. 10%) [44]. Even though prophylaxis has been approved for alloSCT patients, some breakthrough fungal infections have raised attention [45].

### 2.4. Chimeric Antigen Receptor T-Cell Therapy

Data are still scarce on the infections occurring in the course of the most recent cell therapy improvement via chimeric antigen receptor T-cells (CAR T-cells) [46]. Their expected impact on immunity relies on the lympho-depletive chemotherapy preceding the CAR T-cell infusion, the anti-inflammatory drugs used to counter cell-mediated toxicity (tocilizumab against cytokine release syndrome and dexamethasone against neurological complications), the on-target humoral defect induced in the treatment of B-cell malignancies and the often observed persistent or biphasic neutropenia. Fungal infections seem to be rare. In a cohort of 133 patients, 43 patients had presented with an infection at day 28, among which 3% were fungal infections [47]. One of them died from a hemoptysis due to *A. ustus*, being the only *Aspergillus*-documented diagnosis of the cohort [48]. Two probable IA occurred in another series of 53 patients [49].

### 2.5. Acquired or Inherited Immunodeficiency

Although Pneumocystis is the most frequent pathogen in AIDS-associated lung infections, *Aspergillus* is also a classic involved pathogen, ranking second in frequency in a recent Indian study [50]. Incidence could be as high as 19 cases per 100 person-year in reports from the 1990s but have decreased with the improvement of antiretroviral therapy, confirming how CD4 cells count play a major role in the risk [51]. Other factors are neutropenia, corticosteroids, hematological malignancies, lung disease and diabetes [52].

The best characterized inherited deficiency of neutrophilic function is chronic granulomatous disease, in which the production of superoxide anions is impaired by a mutation that can be transmitted either in an autosomal recessive manner or in a X-linked manner. It is a rare disease affecting one patient in 250,000 [53]. These patients usually have a normal count of neutrophils but present, as a result of their dysfunction, with granulomatous lesions and a high risk of staphylococcal and fungal infections. Prophylaxis is strongly recommended as incidence of IA is estimated between 20 and 40% [54].

Polymorphisms in innate immunity mediators have also been shown to increase the risk for IA. They can involve many proteins such as TLR4, TLR1, TLR6, dectin 1, mannose binding lectin, DC-SIGN, IL-1, IL-10, plasminogen or TNF-alpha receptor [55,56,57,58,59].

### 2.6. Solid Organ Transplantations

The 1-year cumulative rate of IA for all types of transplantation was estimated at 0.65% in the data from the TRANSNET study [60]. IA represented 19% of all fungal infections, with a median delay to transplantation of 184 days. IA plays a particular role in lung transplantations, since the pathology setting indication for the transplantation is often associated with *Aspergillus* colonization, as in cystic fibrosis or COPD. This results in a specific pattern of infections, including bronchial anastomotic localizations and a higher frequency of tracheobronchial aspergillosis [61]. Pre-emptive therapy based on positive culture or detection of galactomannan in posttransplant bronchoalveolar lavage fluid reduces the burden of the infection [62]. Various prophylaxis strategies are also used [63,64,65,66].

### 2.7. Solid Tumors

Even though lower than in hematological malignancies, the risk of IA is not to be neglected in solid tumor patients. A study on 452 patients shows how lung cancers, head and neck cancers, gastrointestinal cancers and breast cancers are predominant among the types of tumors involved, suggesting a role of underlying respiratory tract disease, lung irradiation and use of corticosteroids [67]. Focusing on lung cancer patients, the rate of IA reaches 2.6% in a Chinese series [68].

For solid tumors too, therapeutic armamentarium is evolving and brings new kind of risks for fungal infections. Gefitinib (epidermal growth factor receptor tyrosine kinase inhibitor), bevacizumab (anti-vascular endothelial growth factor monoclonal antibody), and temsirolimus (inhibitor of m-TOR) have been considered as potential triggers for IA in case reports [69,70,71,72]. Checkpoint inhibitors (anti-PD-1 receptor monoclonal antibody, e.g., pembrolizumab and nivolumab; anti-CTLA-4 receptor monoclonal antibody, e.g., ipilimumab) do not seem to be responsible for increased IA risk per se [73,74,75,76]. Checkpoint inhibition results in a T-cell activation that might be beneficial in infectious diseases. However, checkpoint inhibition also results in autoimmune adverse events that are treated with molecules impairing host defenses, such as corticosteroids and anti-TNF-alpha inhibitors.

### 2.8. Critically Ill Patients

Overall incidences of 0.3–19% of IA have been reported in intensive care unit (ICU) patients [77]. Some of these patients have other conditions predisposing to IA, such as hematological malignancy or transplant, but depending on the setting of the considered ICU, a high proportion of patients can lack the classical risk factors. For instance, a large analysis of 1850 patients in intensive care unit between 2000 and 2003 has shown an incidence of 6.9% of IA, among which 70% had no hematological malignancy [78]. Non-traditional risk factors met in ICU include systemic corticosteroid use, underlying respiratory diseases, cardiovascular disease, diabetes mellitus and viral pneumonia that will be addressed infra [79]. Outside these comorbidities, a phenomenon of immune paralysis can be induced by severe sepsis and result in a risk factor for IA [80].

More recently, liver failure has been associated with a high risk of fungal infections. A first series of 72 patients presenting with liver failure and IA stresses the role of advanced cirrhosis and corticosteroids [81]. Mortality was high, reaching 71.6%. Acute alcoholic liver failure with high severity score has also been suggested as a risk factor for IA [82].

### 2.9. Viral Pneumonias

IA is an emerging co-infection in patients with influenza who are admitted to the ICU [83]. Influenza-associated pulmonary aspergillosis (IAPA) occurs in ICU patients, of which 57% have no classical risk factor. The virus is responsible for a suppression of NADPH oxidase resulting in dysfunction of neutrophils [84]. Median delay between ICU admission and IAPA diagnosis is 2 to 3 days [85]. ICU mortality reaches 45% in IAPA, compared with 20% for ICU influenza alone.

More recently and with a dramatic accrual into observational studies, IA has been found to be a possible complication of COVID-19 pneumonia. Severe acute respiratory syndrome coronavirus 2 causes direct damages to the airway epithelium that are thought to enable the invasion by *Aspergillus* [86]. In a retrospective multicentric study, among 509 COVID-19 patients, 76 COVID-associated invasive aspergillosis (CAPA) were identified [87]. Risk factors were age over 62, use of dexamethasone and anti-IL6, and duration of mechanical ventilation exceeding 14 days. Death hazard ratio was 1.45 for CAPA patients compared with patients without CAPA. Although the differential diagnosis between colonization and CAPA can be a challenge, awareness is important to allow early antifungal treatment [88,89].

### 2.10. Inflammatory and Autoimmune Diseases

Treatment of inflammatory and autoimmune diseases often requires the use of corticosteroids and immunosuppressive molecules, creating a risk for fungal infections and notably IA. Aside from corticosteroids whose risk is well-known, infliximab (anti-TNF monoclonal antibody) or etanercept (anti-TNF fusion protein) lead to a demonstrated increased risk [90]. Risk may be increased in patients receiving rituximab (anti-CD20 monoclonal antibody), adalimumab (anti-TNF monoclonal antibody) or abatacept (anti-CD28 fusion protein), as well as tocilizumab [91,92,93].

### 2.11. Environmental Factors

Various environmental conditions have been identified as potential risk factors of IA. These factors include notably construction work, geoclimatic factors, use of tobacco or cannabis, contamination of air, food or spices, gardening activity or occupation [3,4,5,94,95,96,97].

## 3. Clinical and Radiological Features

### 3.1. Clinical Features

Symptoms of IA are not specific and it is therefore recommended to adjust the clinician’s awareness according to the risk level of their patients in order to prescribe prompt and relevant further investigations. Fever refractory to, or recrudescent despite, antibiotics is often the first and sometimes the only symptom of IA. Although frequent, fever can be missed in some patients, for instance in those receiving corticosteroids. Cough, sputum production, hemoptysis, pleuritic chest pain or rub, dyspnea or bronchospasm are associated with pulmonary localization of IA.

### 3.2. Radiological Features

Chest radiograph is an obsolete procedure whenever chest computed tomographic (CT scan) is available [98,99]. Contrast agents are usually not necessary for diagnosis but can be useful in assessing the risk of hemoptysis if a lesion is close to a large vessel. Depending on its mechanism of extension, IA can present with two different CT-scan patterns: angio-invasive pattern and airway-invasive pattern [100].

The angio-invasive pattern is the radiological translation of the penetration of hyphae through the vessels wall, leading to fungal thrombi causing necrosis and hematogenous dissemination (Figure 1). It has been classically associated with severe neutropenia and consists of nodules often surrounded by ground-glass opacity, translating perilesional hemorrhage and forming of the halo sign [101]. The halo sign is considered highly suggestive of neutropenia-related IA, but it is not specific and can be seen is various conditions such as other IMD, bacterial infections, primary or metastatic cancers, granulomatous with polyangiitis, bronchiolitis obliterans with organizing pneumonia, or focal lung injury. It is also important to take into account the potential variations of this typical lesion through time: an early CT-scan might show micronodules, whereas a late CT-scan might miss the halo of ground-glass opacity, that classically disappears after 5 to 10 days [102]. In an even later setting, the recovery from neutropenia induces a necrotic retraction of this primary lesion and gives the aspect of an air crescent in a nodule. This air crescent can eventually widen into a cavity, first thick-walled and finally thin-walled.

The airway-invasive pattern is mostly seen in non-neutropenic patients [100,102]. Its CT-scan reflects the endobronchial dissemination of infection with thickened bronchial walls with multiple centrilobular nodules, resulting in the typical tree-in-bud pattern (Figure 2). This pattern is widely shared by other pulmonary infections, from mycobacteria to viruses and is therefore not a strong argument for IA if no other documentation is brought, all the more so as coinfections are frequent. This pattern is also associated with a worse functional course with mechanical ventilation requirement [103].

Both patterns can be intricated in the same patient, showing how complex the IA evolution can be [100]. Moreover, some patients present with only area of consolidation or ground glass opacities [104,105]. Therefore, whether the CT-scan shows a very specific or a non-specific aspect, further investigations must be performed in search for a mycological criterion for IA. CT-scan is also useful for treatment evaluation.

**Figure 1 jof-09-00131-f001:**
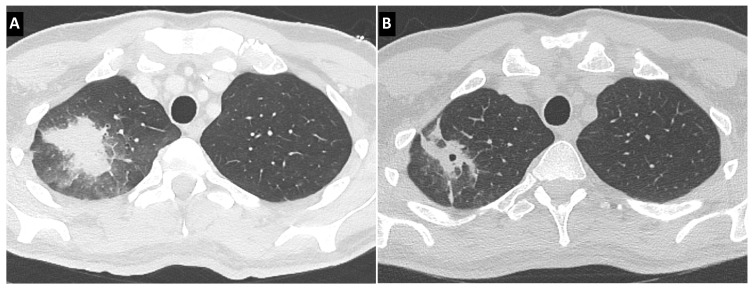
Invasive aspergillosis in a refractory acute myeloblastic leukemia patient. (**A**) CT-scan at diagnosis, showing a nodule surrounded by a halo. (**B**) CT-scan 4 weeks later, showing a small cavity.

**Figure 2 jof-09-00131-f002:**
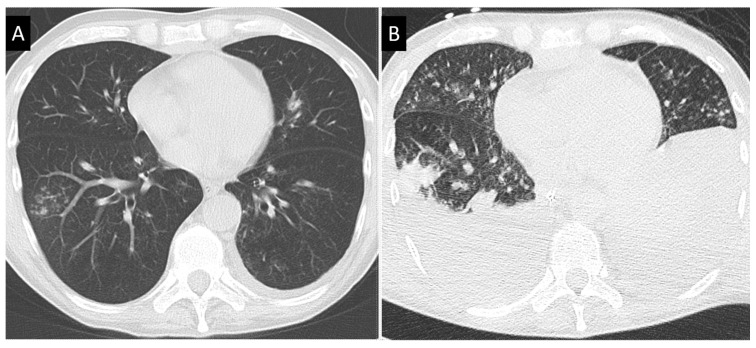
Invasive aspergillosis in a non-neutropenic patient treated for B-cell lymphoma with alemtuzumab and dexamethasone, in a context of PCR-positive influenza A. (**A**) Differential diagnosis is difficult between viral or fungal infection for this tree-in-bud pattern. (**B**) Evolution observed on CT-scan 2 weeks later is characterized by increased size of nodules and pleural effusion. Invasive aspergillosis was eventually documented by repeated detection of galactomannan in serum and galactomannan, as well as septate hyphae, in bronchoalveolar lavage fluid.

Positron-emission tomography coupled with CT-scan (PET-CT) might be an interesting additional part of diagnostic work-up [106,107]. The analysis of fluorodeoxyglucose (FDG) uptake helps better evaluation of the extent of the infection (Figure 3). Further on, PET-CT could be a criterion for treatment discontinuation, as it differentiates active lesions from sequelae [108].

### 3.3. Bronchoscopic Features

Guided by CT-scan findings, a bronchoscopy must be performed whenever the patient is in adequate condition (i.e., without severe hypoxia or bleeding) to help better establish the diagnosis through a bronchoalveolar lavage (BAL) [98]. A higher yield has been obtained in SCT recipients with new pulmonary infiltrates when bronchoscopy was performed within the first 4 days of presentation [109]. Saline is usually used for lavage in the segmental or subsegmental bronchus of the most suggestive area identified on the CT-scan. BAL fluid (BALF) must be assessed by direct examination for cytology, Gram and fungal staining. Bacterial and fungal culture must be performed, as well as mycobacterial if relevant. Biomarkers and polymerase chain reaction (PCR) should be assessed as available and clinically relevant: galactomannan, (1,3)-β-D-glucan (BDG), *Aspergillus* and Mucorales PCR, *Pneumocystis jirovecii* or Toxoplasma PCR, viral PCR. BAL has a pivotal role in detecting polymicrobial infections, the frequency of which can reach 20% in patients with hematological malignancies and should not be neglected [110,111].

Bronchoscopy can also identify the specific and relatively rare feature invasive tracheobronchial aspergillosis (ITBA), which is seen in 7% of intrathoracic aspergillosis [112]. ITBA can present with inflammation and excessive mucus production or with ulcerations and plaque-like lesions, or as a pseudo-membranous necrosis with extensive involvement of the lower airways. It is associated with AIDS, with radiotherapy, or with lung transplantation in a very specific feature just distal to the bronchial anastomosis, compromising the scar [113]. Bronchoscopy helps assessment of the extent of these lesions, poorly defined by CT-scan.

## 4. Microbiological Findings

### 4.1. Samples

Respiratory samples are diverse and not equal in terms of sensitivity and specificity. While the gold-standard for the diagnosis of IA is bronchoalveolar lavage fluid (BALF), its availability relies on the patient’s fitness and willingness for a not innocuous procedure. Sputum is usually more easily available, but a good quality sample can require physiotherapist maneuvers for patients with a little-productive cough. In intubated patients, tracheal aspirates or brushes are an interesting intermediate sample. However, both sputum and tracheal aspirates may lead to a confusion between infection and colonization if not interpreted with caution.

Culture-positive blood-samples are exceptional, in spite of the angio-invasive capacity of *Aspergillus* species [114]. Nevertheless, blood samples are part of the diagnostic work-up of IA, in the search for biomarkers and fungal DNA. According to manufacturer specifications, tests can be performed on whole blood, on plasma or serum. Thresholds for positivity may vary depending on the nature of the sample.

### 4.2. Direct Microscopy

Microscopic examination should be performed on all the available respiratory samples (BALF, tracheal aspiration or sputum), in spite of its low yield. Performance can be enhanced by the use of calcofluor-white [115]. *Aspergillus* is identified as a septate hyphae with dichotomous acute angle branching.

### 4.3. Culture

Commonly used media for *Aspergillus* culture are Sabouraud-dextrose, brain-heart-infusion or potato-dextrose agar [115]. Chloramphenicol is used to avoid competition with bacteria, while cycloheximide can be used to reduce environmental fungal contamination but might reduce the yield. Additional specialized media can be used to select *Aspergillus* species. Culture should be incubated at 30 °C for 21 days in a humidified environment. Mass spectrometry identification (MALDI-TOF) or genomic sequencing can be used on positive culture if microscopy does not enable a species identification [10,116]. Even more importantly, a positive culture enables testing of the strain’s sensitivity to antifungals.

### 4.4. Antibody

As IA is associated with immunosuppression, the lack of detection of antibodies directed against *Aspergillus* cannot be used as a diagnostic argument, contrary to the setting of ABPA or CPA [117]. Some patients still present with a humoral response leading to the detection of such antibodies within a mean time of 10 days after the onset of IA. The presence of anti-*Aspergillus* antibodies may decrease the sensitivity of the galactomannan detection test [118].

### 4.5. Galactomannan

GM is a polysaccharide component of the fungal cell wall of *Aspergillus*. GM detection is validated in serum, BALF and cerebrospinal fluid, with a better sensitivity than culture, and feasible but not validated in sputum and tissue biopsies. The results are expressed as an index in comparison to a control. The cut-off for positivity in serum is still controversial: single value over 0.5 is recommended by the manufacturer, while experts suggest a cut-off over 0.7 for a single test or two consecutive tests > 0.5 [119]. In BALF too, there is no consensus between >0.5 and >1.0, although a recent meta-analysis including 19 articles concluded in favor of the >0.5 cut-off associated with a sensitivity of 89% and a specificity of 79% [120].

Its high sensitivity has made GM detection an important part of diagnostic work-up for suspected IA in neutropenic patients. It can reach 60 to 80% in hematology patients but is lower in non-neutropenic patients and poor in the setting of anti-mold prophylaxis [118,121,122]. GM can be considered as very specific, but some cross reactivity can occur with other molds or dimorphic fungi such as *Fusarium* sp., *Alternaria* sp., *Acremonium* sp., *Penicillium* sp., *Paecylomyces* sp., *Wangiella dermatitidis*, *Histoplasma capsulatum* or *Blastomyces dermatitidis* [122]. Use of beta-lactam antibiotics or severe mucosal lesions including GVHD have led to false positive tests in serum, while some saline solutions used in BAL have led to false positive tests in BALF.

When positive, an early decrease of serum GM index has been associated with a higher favorable response rate [123].

### 4.6. (1,3)-β-D-Glucan

Detection of BDG in serum might add sensitivity to the diagnostic work-up [124,125]. However, as this glucan is a component of the cell wall of most fungi, including *Aspergillus* but also *Candida* and *Pneumocystis*, its usefulness is limited by a lack of specificity. As BDG is not produced by *Mucorales* and *Cryptococcus*, it can still be of help in differential diagnosis. BDG can be detected also in BALF, but its poor specificity once again limits its usefulness [126].

### 4.7. Polymerase Chain Reaction

*Aspergillus* PCR has long lacked availability and standardization and is therefore less developed than GM use. These obstacles have now been overcome and PCR is included in routine diagnostic work-up with commercial kits based on serum, plasma or BALF samples [127,128,129,130,131]. Other samples can be analyzed, such as fresh tissue or formalin fixed and paraffin wax embedded tissue [132].

PCR sensitivity is high and can reach >90% in routine blood samples [133]. Its combination with GM detection results in BALF has been suggested to reach 96% sensitivity [134]. PCR specificity is, as expected, very high, up to 100% in some studies [135]. Cross-reactivities have nevertheless been reported with other molds such as *Penicillium* spp., *Fusarium* spp. and *Rhizopus oryzae* [136].

An additional value of PCR is the possibility of using probes against resistance-associated mutations, for the detection of azole-resistance, for instance [137].

### 4.8. Point of Care Tests

Point of care tests, also known as “bedside testing,” are particularly useful for rapid start to relevant antifungal therapy. Two of them available and European conformity CE-marked: lateral flow device AspLFD (OLM Diagnostics, Newcastle-on-Tyne, UK) detecting an extracellular glycoprotein associated with *Aspergillus* growth, and lateral flow assay (IMMY, Norman, OK, USA) detecting GM [138,139]. Sensitivity and specificity are over 70% [140,141].

## 5. Diagnostic Criteria

### 5.1. EORTC-MSG Criteria

Confronted with the heterogeneity of features of IA reported in international literature, experts of the EORTC-MSG (European Organization for the Research and Treatment of cancer/Mycosis Study Group) issued as early as 2002 a system of criteria enabling determination of three levels of diagnostic certainty [142]. This system was not intended for clinical purpose and did not aim at refraining the start of a treatment due to lack of a criterion. The intent is harmonization of the clinical trials for a better understanding of their results. To account for improvements in diagnostic tools and growing variability of radiological findings associated with IA, updates have been issued in 2008 and 2020 [143,144]. This review will focus on the latter.

An IA is considered to be proven in the case of a positive mycology or histopathology of a sterile sample obtained in a sterile manner (Figure 4). This is restricted to needle aspirations and biopsies as opposed to BAL fluid. Positive mycology can consist of a positive culture, PCR or DNA sequencing. Positive histopathology consists of a sample in which hyphae are seen accompanied by evidence of tissue damage. If an identification of the mold is not performed, the diagnosis will be of IMD rather than IA.

Other levels of diagnostics levels of IA require a more thorough examination of the whole patient’s situation [144]. The host criterion relies on the presence of a known risk factor among the following: recent neutropenia (neutrophils < 0.500 G/L for more than 10 days), active hematological malignancy, alloSCT, solid organ transplantation, prolonged use of corticosteroids at a minimum dose of 0.3 mg/kg/d of prednisone or equivalent for more than 3 weeks, acute GvHD, T-cell suppressants, B-cell suppressants, inherited severe immunodeficiency (Figure 1). The clinical criterion for pulmonary aspergillosis is met in case of dense, well-circumscribed lesion with or without a halo sign, or an air crescent sign, or a cavity, or a wedge-shaped and segmental or lobar consolidation, whereas the clinical criterion for tracheobronchitis requires a tracheobronchial ulceration, nodule, pseudo-membrane, plaque or eschar seen on bronchoscopy. The mycological criterion is met in case of a positive direct microscopy or culture in sputum, BALF, bronchial brush or aspirate, or GM detected in blood ≥ 1.0, or in BALF ≥ 1.0 or both blood ≥ 0.7 and BALF ≥ 0.8, or a PCR positive in blood twice or in BALF twice (first analysis and duplicate) or once in both blood and BALF.

The diagnosis is probable IA if host criterion, clinical criterion and mycological criterion are all met. It is possible IMD if only host and clinical criterion are met (Figure 5).

### 5.2. AspICU

In spite of their focus-widening updates, the EORTC-MSG criteria remain restrictive with regard to a large category of patients susceptible to developing IA, particularly in critical care setting. Therefore, parallel diagnostic criteria were developed more adapted to critically ill patients, the Aspergillosis Intensive Care Unit algorithm, known as AspICU (Figure 5) [145]. The proven IA category shares the definition of the EORTC-MSG proven IA. The IA is considered putative in cases in which four criteria are met: the entry criterion is *Aspergillus*-positive lower respiratory tract specimen culture; the clinical criterion comprises refractory or recrudescent fever despite antibiotics, pleuritic chest pain or rub, hemoptysis, dyspnea or respiratory insufficiency; the radiological criterion consists of an abnormal medical imaging by portable chest X-ray or CT scan of the lungs; the fourth criterion can be either a host risk factor (neutropenia, cytotoxic-treated hematological or oncological malignancy, glucocorticoid treatment or immunodeficiency) or a mycological finding: semiquantitative *Aspergillus*-positive culture of BALF without bacterial growth together with a positive cytological smear showing branching hyphae. If any of the four criteria is lacking, the case is considered a colonization.

To address the data provided by biomarkers, a new algorithm, BM-AspICU, was proposed [146]. The entry criterion is either positive *Aspergillus* in the lower respiratory tract, a radiological sign, or a clinical sign such as described in AspICU (Figure 5). If the patient has a “strong” host factor as defined by EORTC-MSG, a combination of an abnormal imaging and a mycological criterion are sufficient to diagnose a probable IA. If the patient only has a “weak” host factor as defined in AspICU, the combination of a clinical, a radiological and two mycological criteria are needed for the diagnosis of probable IA. Otherwise, the diagnosis is colonization.

### 5.3. Invasive Aspergillosis in Specific Conditions

More and more often described as a specific entity, influenza-associated pulmonary aspergillosis (IAPA) has its own consensual algorithm for diagnosis, associating pulmonary infiltrates in the setting of PCR-positive influenza with either tracheobronchitis, GM in serum or BALF, culture-positive BALF, or culture-positive sputum associated with an image of cavitation, to draw the diagnosis of probable IAPA [147].

With the high prevalence of *Aspergillus*-positive samples in COVID19-patients, a consensus for COVID19-associated invasive aspergillosis (CAPA) had to be found in the literature documenting the epidemy. A system of clinical, radiological and mycological criteria was proposed [86]. The clinical criteria are similar to that of AspICU: refractory fever, pain, pleuritic rub, dyspnea or hemoptysis. The radiological criterion consists of a pulmonary infiltrate or a cavity. The association of both with a strong mycological criterion (microscopy, culture, GM or PCR positive in BALF) leads to the diagnosis of probable CAPA. Their association with a weak mycological criterion (microscopy, culture, GM or PCR positive in bronchial aspiration) prompts the diagnosis of possible CAPA.

## 6. Treatment

Even outside the diagnostic criteria described supra, which were meant for clinical research purposes, one should consider the start of an antifungal treatment if the diagnosis of IA seems consistent with the clinical, radiological and, if available, mycological, findings.

### 6.1. Amphotericin B and Lipid Formulations

The first antifungal to show activity against *Aspergillus* was historically the polyene deoxycholate amphotericin B, whose poor safety profile has now led to recommendation against its use [17,148]. It is nevertheless still useful in low-resource settings [149].

An important improvement came from lipid formulations of amphotericin B, i.e., liposomal amphotericin B (L-amB), amphotericin B lipid complex (ABLC) and colloidal dispersion (ABCD). All were associated with less nephrotoxicity, and none with lower efficacy than deoxycholate amphotericin B [150,151]. Exhibiting the best safety-profile, L-amB has eventually become the most widely available and used among the lipid formulations of amphotericin B, with a documented efficacy against IA [16]. The usual dosage against *Aspergillus* is 3 mg/kg/day and attempts to enhance antifungal activity with higher dosage in highly immunocompromised patients have proven disappointing. L-amB is particularly useful in the setting of a breakthrough infection under azole prophylaxis or in salvage therapy [139]. Aerosolization of amphotericin B and L-amB can also be used as a prophylaxis for IA, the latter showing a better tolerance [152,153,154].

### 6.2. Azoles

Inhibiting the synthesis of membrane component ergosterol, azoles play a key role in the management of many fungal infections [155]. Fluconazole lacking any efficacy on *Aspergillus*, the first azole used in the treatment of IA was itraconazole, with 39% patients having a complete or partial response at the end of treatment but a limiting toxicity and concerns about bioavailability [156].

A major step in IA treatment was made with the development of voriconazole. In the pivotal randomized trial comparing voriconazole to amphotericin B in IA treatment, successful outcomes reached 52.8% vs. 31.6% [17,157]. Tolerability of voriconazole is generally good, with a restriction due to frequent liver toxicity, visual and neurological side-effects and an interaction profile that may lead to difficulties in managing co-administration with hematological drugs such as cyclosporine, midostaurine or venetoclax. Rarely, prolonged use can lead to phototoxicity, skin carcinomas, peripheral neuropathy and periostitis [158]. Therapeutic-drug monitoring is available and should be used. The advised dosage is 6 mg/kg bid for a charging period of 24 h followed by 4 mg/kg bid. Voriconazole is recommended as a first-line treatment by IDSA, ECIL and ESCMID [98,99,148].

Posaconazole was first used in second line treatment for patients refractory to or intolerant of conventional antifungal therapy, with a satisfactory success rate of 42% [159]. In terms of tolerance, posaconazole exhibits a better safety profile than voriconazole [160]. This tolerability has allowed usage as a prophylaxis, all the more so since delayed-release tablets have enhanced bioavailability that used to be a concern [161]. Another interesting advantage of posaconazole upon voriconazole is its activity against Mucorales [162,163], for which differential diagnosis with IA can be challenging. Direct comparison between voriconazole and posaconazole in the treatment of IA has recently been achieved and shows the absence of inferiority of the latter, with a better safety-profile [160]. Recommended dosage is 300 mg bid for 1 day, followed by 300 mg qod and therapeutic drug monitoring with a target trough of 0.7 mg/L for prophylaxis or 1.0 mg/L for treatment.

Isavuconazole is the latest approved azole and shows satisfactory activity against *Aspergillus*, combined with an excellent safety profile. It has been compared with voriconazole in a randomized double-blind study [164], that demonstrated non-inferiority in terms of efficacy with a similar survival at week 6 and week 12. It was better tolerated with less hepatic, visual and cutaneous adverse events, and can therefore be of particular interest in the treatment of patients at risk of hepatic impairment. Moreover, its interaction profile with various drugs is more easily managed and leads to less dosage adaptations of concomitant therapies. A meta-analysis of isavuconazole trials concluded that its efficacy is comparable to both voriconazole and L-amB [165]. Besides, isavuconazole shows an interesting activity against Mucorales [166].

### 6.3. Echinocandins

Echinocandins are BDG-synthesis inhibitors that show mitigated activity against voriconazole [167]. Caspofungin was first used as a second line therapy for IA patients with intolerance or failure of previous therapy and led to an encouraging 45% of favorable responses [168]. In first line therapy, it has been tried in neutropenic patients and alloSCT patients with results below expectations (53% and 50% survival at week 12, respectively), and is therefore not recommended by guidelines [169,170]. Micafungin and anidulafungin have insufficient activity against *Aspergillus* to be used as treatment in IA. On the other hand, the upcoming rezafungin shows promising activity against *Aspergillus* in vitro and mouse models [171].

### 6.4. Perspectives

Other new antifungals are awaited [172]. Ibrexafungerp, a BDG-synthesis inhibitor, is currently being tried in combination with voriconazole against voriconazole monotherapy in the setting of IA (NCT03672292; https://clinicaltrials.gov/, accessed on 22 January 2022) after promising results in vitro and mouse models in combination with isavuconazole [173,174].

Olorofim, an inhibitor of dihydro-oroate dehydrogenase, is currently being tried against L-amB (NCT05101187; https://clinicaltrials.gov/ accessed on 22 January 2022) in the setting of IA after promising results in vitro and in mouse models of neutropenia [175,176].

Fosmanogepix (APX001), an inhibitor of the fungal enzyme Gwt1, is currently in phase 2 trial (NCT04240886; https://clinicaltrials.gov/ accessed on 22 January 2022) after promising preclinical results [177].

Opelconazole is a new triazole designed for inhalation with a sustained lung residency which might be of use for prophylaxis or treatment of IA [172].

### 6.5. Combination Therapies

Several publications have explored the efficacy of combination therapy: L-amB and caspofungin, voriconazole and caspofungin or voriconazole and anidulafungin [178,179,180,181,182]. Only the latter was a prospective randomized large-scale study and no superiority of the combination therapy arm was demonstrated. Therefore, guidelines do not recommend the use of two antifungals for treating IA [98,99,148]. However, combination can be argued for in the context of high azole-resistance prevalence and might be of use in salvage therapy.

### 6.6. Antifungal Strategies

The availability of many different antifungals with various efficacy and safety profiles prompts the establishment of management strategies (Figure 5 and Figure 6) [183]. The relevance of anti-mold prophylaxis for patients at risk after AML-remission induction therapy or during alloSCT procedure must be discussed on the basis of the local epidemiology [21]. Other situations that can also be an indication for prophylaxis are not as consensual as yet.

Empirical therapy, taking place at the start of antifungal agents in any antibacterial-resistant fever for at-risk neutropenic patients, was proposed four decades ago and decreased morbidity and mortality. However, following studies had conflicting results [184,185]. Concerns are raised about exposing patients to useless toxicities, promoting resistance, and about overall costs.

Preemptive therapy, based on a diagnostic-driven start of antifungals, was made possible through the improvement of efficiency and availability of biomarkers and CT-scan. It has been shown to lessen the consumption of antifungals without impairing patients’ outcome [18,186].

Whichever way the start of treatment is decided, every clinician is faced with the decision regarding the discontinuation of antifungal therapy. Standard recommendations point towards 6 to 12 weeks, but the treatment duration should be tailored to the patient’s situation [128]. Response assessment with CT-scan, or perhaps PET-CT, can help in the decision-making. If the patient remains at-risk, treatment is often switched into secondary prophylaxis, sometimes with the same antifungal.

**Figure 5 jof-09-00131-f005:**
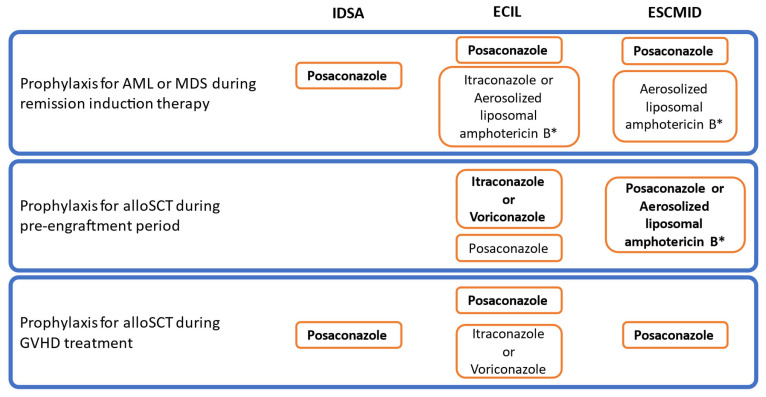
Recommendations from the IDSA, ECIL and ESCMID for prophylaxis of invasive aspergillosis [21,99,128]. Abbreviations: IDSA: Infectious Disease Society of America; ECIL: European Conference on Infections in Leukemia; ESCMID: European Society of Clinical Microbiology and Infectious Diseases; AML: acute myeloid leukemia; MDS: myelodysplastic syndrome; alloSCT: allogeneic stem cell transplantation; GVHD: graft vs. host disease. * Fluconazole should be added for coverage of yeast infections in case aerosolized liposomal amphotericin B is used as an anti-mold prophylaxis. Antifungals are **written in bold font** when the recommendation is first choice, based on strong-quality evidence.

An algorithm has been suggested to decide when to stop antifungal treatment in patients with hematological malignancies and IA. It integrates status of the hematological malignancy, recovery from neutropenia, negative mycology, clinical and imaging response to antifungal therapy and planned further chemotherapy and immunosuppression [108].

**Figure 6 jof-09-00131-f006:**
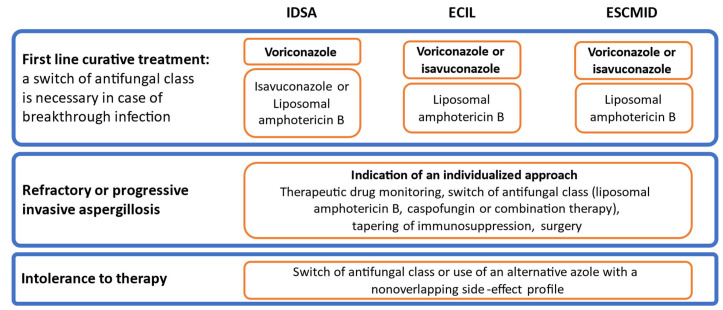
Recommendations from the IDSA, ECIL and ESCMID for targeted treatment of IA [99,128,148]. Abbreviations: IDSA: Infectious Disease Society of America; ECIL: European Conference on Infections in Leukemia; ESCMID: European Society of Clinical Microbiology and Infectious Diseases; AML: acute myeloid leukemia; MDS: myelodysplastic syndrome; alloSCT: allogeneic stem cell transplantation; GVHD: graft vs. host disease. Antifungals are **written in bold font** when the recommendation is first choice, based on strong-quality evidence.

## 7. Conclusions

IA, a risk for more and more patients in cancer and critical care, remains a difficult diagnosis but improvement has been made through better imaging and many available biomarkers. Treatment has seen a decisive widening of the armamentarium and further improvement is still expected. An abundant literature and many adapted guidelines are available to help the clinician through diagnosis and management of IA patients.

## Figures and Tables

**Figure 3 jof-09-00131-f003:**
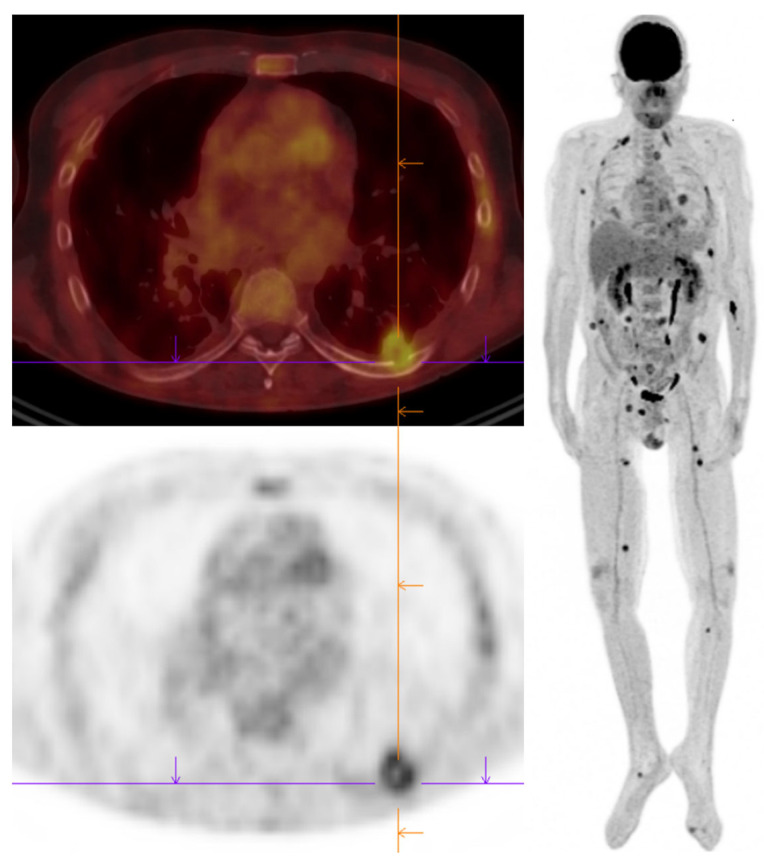
Positron-emission tomography coupled with CT-scan in invasive aspergillosis in a solid organ transplant patient. Primary localization is pulmonary with an extent to chest wall. Dissemination to multiple organs including heart and soft tissue were diagnosed on PET-CT.

**Figure 4 jof-09-00131-f004:**
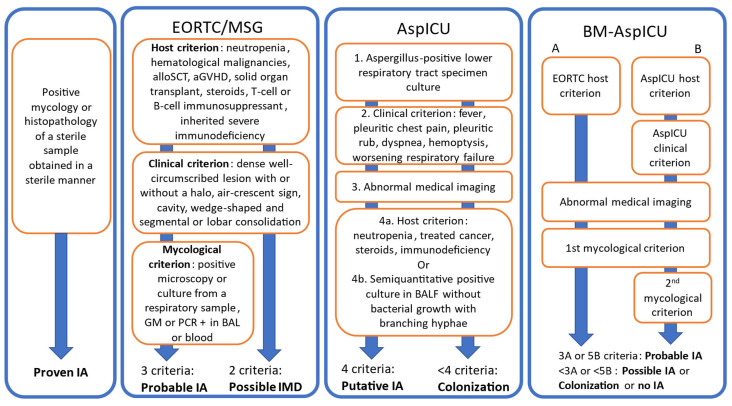
Invasive pulmonary aspergillosis diagnostic criteria [144,145,146]. Abbreviations: EORTC-MSG: European Organization for Research and Treatment of Cancer/Mycosis Study Group; AspICU: Aspergillosis in Intensive Care Unit diagnostic algorithm; BM-AspICU: Biomarkers-based Aspergillosis in Intensive Care Unit diagnostic algorithm; AlloSCT: allogeneic stem cell transplantation; aGVHD: acute graft vs. host disease; GM: galactomannan; PCR: polymerase chain reaction; BALF: bronchoalveolar lavage fluid, IA: invasive aspergillosis; IMD: invasive mold infection. EORTC-MSG 2020 mycological criterion consists of. -Direct microscopy or culture positive in sputum, bronchoalveolar lavage fluid (BALF), bronchial brush or aspirate, or. -GM index >1 in blood sample or >1 in BALF or >0.7 in blood sample and >0.8 in BALF, or. -PCR positive twice in blood or twice in BALF or once both in blood and BALF. BM-AspICU mycological criterion consists of positive culture in BALF, or positive GM or PCR in blood or BALF.

## Data Availability

Not applicable.

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
