# Peer review of "Invasive Pulmonary Aspergillosis"

_jof, 2023, doi:10.3390/jof9020131_

Round 1

Reviewer 1 Report

J of F 2104714  ASPERGILLOSIS

 This is an excellent review written by leading authorities on the subject

I have some rather minor points for the authors to consider for their revision

L 196. Check point inhibitors ( CPIs) per se are NOT associated with increased IA risk. In contrast, treatment of autoimmune side effects of CPIs by corticosteroids and or TNF inhibitors (infliximab) does result in some excess IA, even in patients at low risk for invasive aspergillosis such as patients with melanoma.(see  The Spectrum of Serious Infections Among Patients Receiving Immune Checkpoint Blockade for the Treatment of Melanoma. Del Castillo M, Romero FA, Argüello E, Kyi C, Postow MA, Redelman-Sidi G.Del Castillo M, et al. Clin Infect Dis. 2016 Dec 1;63(11):1490-1493. doi: 10.1093/cid/ciw539. Epub 2016 Aug 7.Clin Infect Dis. 2016. PMID: 27501841 , .  Checkpoint Inhibition and Infectious Diseases: A Good Thing? Abers MS, Lionakis MS, Kontoyiannis DP.. Trends Mol Med. 2019 Dec;25(12):1080-1093. doi: 10.1016/j.molmed.2019.08.004..Trends Mol Med. 2019. PMID: 31494023

 L242: recent studies implicate marijuana use as risk factor for IPA (Emerg Infect Dis

 2020 Jun;26(6):1308-1310. doi: 10.3201/eid2606.191570. Cannabis Use and Fungal Infections in a Commercially Insured Population, United States, 2016 Kaitlin BenedictGeorge R Thompson 3rdBrendan R Jackson)  

L299. See some algorithmic guidance in How Long Do We Need to Treat an Invasive Mold Disease in Hematology Patients? Factors Influencing Duration of Therapy and Future Questions. Fernández-Cruz A, Lewis RE, Kontoyiannis DP.. Clin Infect Dis. 2020 Jul 27;71(3):685-692. doi: 10.1093/cid/ciz1195.Clin Infect Dis. 2020. PMID: 32170948

 L 305. Is timing of bronchoscopy (early vs late) matters for the yield? Some (old) data suggest it does (Utility of early versus late fiberoptic bronchoscopy in the evaluation of new pulmonary infiltrates following hematopoietic stem cell transplantation. Shannon VR, Andersson BS, Lei X, Champlin RE, Kontoyiannis DP. Bone Marrow Transplant. 2010 Apr;45(4):647-55. doi: 10.1038/bmt.2009.203. Epub 2009 Aug 17.Bone Marrow Transplant. 2010. 0

 L 335. Aspergillus terreus might be an exception, as it is not uncommonly associated with (true) aspergillemia (Clin Infect Dis . 2000 Jul;31(1):188-9. doi: 10.1086/313918. Significance of aspergillemia in patients with cancer: a 10-year study D P Kontoyiannis  1 D SumozaJ TarrandG P BodeyR StoreyI I Raad

 L 382:  See reference that documents the lack of usefulness of b-D glucan in BAL in patients with IPA (J Infect . 2014 Sep;69(3):278-83. doi: 10.1016/j.jinf.2014.04.008. Epub 2014 May 4. The utility of bronchoalveolar lavage beta-D-glucan testing for the diagnosis of invasive fungal infections Stacey R Rose  1 Saraschandra Vallabhajosyula  2 Miguel G Velez  2 Daniel P Fedorko  3 Mark J VanRaden  4 Juan C Gea-Banacloche  5 Michail S Lionakis )

 Consider emphasizing the efficacy the new Aspergillus-active triazoles isavuconazole and voriconazole is suboptimal in neutropenic patients with IA.( Impact of unresolved neutropenia in patients with neutropenia and invasive aspergillosis: a post hoc analysis of the SECURE trial. Kontoyiannis DP, Selleslag D, Mullane K, Cornely OA, Hope W, Lortholary O, Croos-Dabrera R, Lademacher C, Engelhardt M, Patterson TF. J Antimicrob Chemother. 2018 Mar 1;73(3):757-763.)

Author Response

Dear Editor

We thank the reviewers for their comments. They are relevant and we have modified the manuscript accordingly.

Reviewer 1:

  1. L 196. Check point inhibitors ( CPIs) per se are NOT associated with increased IA risk. In contrast, treatment of autoimmune side effects of CPIs by corticosteroids and or TNF inhibitors (infliximab) does result in some excess IA, even in patients at low risk for invasive aspergillosis such as patients with melanoma.(see  The Spectrum of Serious Infections Among Patients Receiving Immune Checkpoint Blockade for the Treatment of Melanoma. Del Castillo M, Romero FA, Argüello E, Kyi C, Postow MA, Redelman-Sidi G.Del Castillo M, et al. Clin Infect Dis. 2016 Dec 1;63(11):1490-1493. doi: 10.1093/cid/ciw539. Epub 2016 Aug 7.Clin Infect Dis. 2016. PMID: 27501841 , .  Checkpoint Inhibition and Infectious Diseases: A Good Thing? Abers MS, Lionakis MS, Kontoyiannis DP. Trends Mol Med. 2019 Dec;25(12):1080-1093. doi: 10.1016/j.molmed.2019.08.004..Trends Mol Med. 2019. PMID: 31494023

Reply: We agree and have modified the manuscript Lines 199 to 205: the impact of checkpoint inhibitors is now analyzed in 3 separate sentences and appropriate references (N° 73 to 76) have been moved here or added.

  1. L242: recent studies implicate marijuana use as risk factor for IPA (Emerg Infect Dis 2020 Jun;26(6):1308-1310. doi: 10.3201/eid2606.191570. Cannabis Use and Fungal Infections in a Commercially Insured Population, United States, 2016 Kaitlin Benedict, George R Thompson 3rd, Brendan R Jackson)

Reply: we acknowledge the reviewer’s comment although this point remains debatable. To further develop the potential role of environmental factors we have added a specific paragraph (2.11) Lines 247 to 251 with additional references.

  1. See some algorithmic guidance in How Long Do We Need to Treat an Invasive Mold Disease in Hematology Patients? Factors Influencing Duration of Therapy and Future Questions. Fernández-Cruz A, Lewis RE, Kontoyiannis DP.. Clin Infect Dis. 2020 Jul 27;71(3):685-692. doi: 10.1093/cid/ciz1195.Clin Infect Dis. 2020. PMID: 32170948

Reply: We have added the reference (N°108) suggested by reviewer in Line 316 and developed the interest of the algorithm mentioned by reviewer in the paragraph dedicated to duration of treatment in Lines 649 to 652 supported by reference (N°108).

  1. L 305. Is timing of bronchoscopy (early vs late) matters for the yield? Some (old) data suggest it does (Utility of early versus late fiberoptic bronchoscopy in the evaluation of new pulmonary infiltrates following hematopoietic stem cell transplantation. Shannon VR, Andersson BS, Lei X, Champlin RE, Kontoyiannis DP. Bone Marrow Transplant. 2010 Apr;45(4):647-55. doi: 10.1038/bmt.2009.203. Epub 2009 Aug 17.Bone Marrow Transplant. 2010. 0

Reply: We have introduced a sentence on the benefit of doing early BAL in Lines 327 to 329 and reference 109

  1. L 335. Aspergillus terreus might be an exception, as it is not uncommonly associated with (true) aspergillemia (Clin Infect Dis . 2000 Jul;31(1):188-9. doi: 10.1086/313918. Significance of aspergillemia in patients with cancer: a 10-year study D P Kontoyiannis 1 , D Sumoza, J Tarrand, G P Bodey, R Storey, I I Raad

Reply: Although we still are convinced Aspergillus fungemia are very rare, we recognize that when fungemia occurs the species involved is most often A. terreus. Therefore we have not changed our sentence but we have added the reference suggested by reviewer in Line 357, reference N° 114.

  1. L 382: See reference that documents the lack of usefulness of b-D glucan in BAL in patients with IPA (J Infect . 2014 Sep;69(3):278-83. doi: 10.1016/j.jinf.2014.04.008. Epub 2014 May 4. The utility of bronchoalveolar lavage beta-D-glucan testing for the diagnosis of invasive fungal infections Stacey R Rose  1 , Saraschandra Vallabhajosyula  2 , Miguel G Velez  2 , Daniel P Fedorko  3 , Mark J VanRaden  4 , Juan C Gea-Banacloche  5 , Michail S Lionakis )

Reply: we fully agree that beta-D-glucan detection test lacks specificity in both serum and BAL fluid. To clarify this point we have added a sentence on beta-D-glucan in BAL in Lines 408 and 409 with an additional reference (N°126)

  1. Consider emphasizing the efficacy the new Aspergillus-active triazoles isavuconazole and voriconazole is suboptimal in neutropenic patients with IA.( Impact of unresolved neutropenia in patients with neutropenia and invasive aspergillosis: a post hoc analysis of the SECURE trial. Kontoyiannis DP, Selleslag D, Mullane K, Cornely OA, Hope W, Lortholary O, Croos-Dabrera R, Lademacher C, Engelhardt M, Patterson TF. J Antimicrob Chemother. 2018 Mar 1;73(3):757-763.)

Reply: It is well accepted that absence of recovery from neutropenia is associated with a poor outcome whatever the antifungal agent and this is already stated in the manuscript in Lines 101 to 103. We do not feel that an additional reference is mandatory.

---------------------------------------------------------------------------------------

Additional changes

In addition to the suggestions of the reviewers we have added a few changes:

  1. Figure 3 appeared pixelated. We have increased the resolution and the new version looks better.
  2. Table 1 :
    1. Titled has been modified with the replacement of “invasive aspergillosis” by “invasive pulmonary aspergillosis” as the AspICU and BM-AspICU criteria only apply to pulmonary infections.
    2. Addition of hematological malignancy and solid organ transplantation in the list of host criteria for the EORTC/MSG definitions. These two factors already appear in the text and we feel they should also be present in the table.
    3. In the lower part of the figure, within the “EORTC-MSG” frame, we replace the terms “possible IA” by “possible IMD” since it is more accurate.
  3. Line 224: the detailed name for IAPA has been corrected into Influenza-associated pulmonary aspergillosis.

Reviewer 2 Report

well written and with excellent, text, Figures, and Tables. The only error detected was on line 377 where "  lactam" should be expanded to "beta lactam". Apart from this change, I recommend acceptance as written.

Author Response

Dear Editor

We thank the reviewers for their comments. They are relevant and we have modified the manuscript accordingly.

Reviewer 2

  1. Well written and with excellent, text, Figures, and Tables. The only error detected was on line 377 where "  lactam" should be expanded to "beta lactam". Apart from this change, I recommend acceptance as written.

Reply: We thank the reviewer for the encouraging comments. Typo in line 398 has been corrected.

----------------------------------------------------------------------------------------

Additional changes

In addition to the suggestions of the reviewers we have added a few changes:

  1. Figure 3 appeared pixelated. We have increased the resolution and the new version looks better.
  2. Table 1 :
    1. Titled has been modified with the replacement of “invasive aspergillosis” by “invasive pulmonary aspergillosis” as the AspICU and BM-AspICU criteria only apply to pulmonary infections.
    2. Addition of hematological malignancy and solid organ transplantation in the list of host criteria for the EORTC/MSG definitions. These two factors already appear in the text and we feel they should also be present in the table.
    3. In the lower part of the figure, within the “EORTC-MSG” frame, we replace the terms “possible IA” by “possible IMD” since it is more accurate.
  3. Line 224: the detailed name for IAPA has been corrected into Influenza-associated pulmonary aspergillosis.